# Effect of Baduanjin exercises on elevated blood lipid levels of middle-aged and elderly individuals: protocol for a systematic review and meta-analysis of randomised controlled trials

Junmao Wen,[1] Tong Lin,[1] Chenguang Jiang,[1] Rui Peng,[2] Wei Wu[2]

[1]Guangzhou University of Chinese Medicine, Guangzhou
[2]Department of cardiovascular, The First Affiliated Hospital of Guangzhou University of Chinese Medicine, Guangzhou, China

**Correspondence to**
Dr. Wei Wu;
3332564768@qq.com

## ABSTRACT

**Objective** To assess the safety and effect on elevated blood lipid levels of Baduanjin exercises in middle-aged and elderly individuals.

**Methods** A systematic literature search for articles up to March 2017 will be conducted using seven databases: PubMed, Embase, Cochrane Library, Chinese National Knowledge Infrastructure, Chinese Science and Technology Periodicals Database, Chinese BioMedical Database and Wanfang Data. Inclusion criteria are randomised controlled trials of Baduanjin exercises that examine blood lipid levels in middle-aged and elderly individuals. The primary outcome measures will be total cholesterol, triglycerides, low-density lipoprotein cholesterol and high-density lipoprotein cholesterol. Stata V.13.0 software will be used for data synthesis, sensitivity analysis, metaregression, subgroup analysis and risk of bias assessment. A funnel plot will be developed to evaluate reporting bias and Begg and Egger tests will be used to assess funnel plot symmetries. We will use the Grading of Recommendations Assessment, Development and Evaluation system to assess the quality of evidence.

**Ethics and dissemination** This systematic review will be submitted to a peer-reviewed journal. Our findings will provide information about the safety of Baduanjin exercises for middle-aged and elderly individuals and their effect on elevated blood lipid levels.

**Trial registration number** PROSPERO CRD 42017060613.

## INTRODUCTION

### Description of the condition

Elevated blood lipid levels are risk factors for problems such as cardiovascular disease, stroke and diabetes.[1 2] Abnormal blood lipid metabolism, characterised by an increase in low-density lipoprotein cholesterol (LDL-C) or total cholesterol (TC), is a critical factor for subsequent atherosclerotic cardiovascular disease.[3 4] Statins, fibrates, nicotinic acid and bile acid sequestrants are effective in regulating blood lipids.[5 6] Statins are the most widely used pharmacological agents.

### Strengths and limitations of this study

► This study will assess the safety of Baduanjin exercises and their effect on elevated blood lipid levels in middle-aged and elderly individuals.

► Two reviewers will independently conduct the data extraction and risk of bias assessment.

► The Grading of Recommendations Assessment, Development and Evaluation system will be used to further evaluate study findings.

► There may be a language bias, as both English and Chinese studies will be included.

► There may be clinical heterogeneity because of variations in treatment frequency and duration and the use of additional therapies (eg, herbal medicine).

They significantly lower TC and LDL-C levels, which reduces cardiovascular events. The role of statins in the primary prevention of cardiovascular diseases in high-risk populations has been recognised.[7] However, statins have adverse effects on muscles, leading to problems such as myosalgia, initis and rhabdomyolysis.[8] Furthermore, the long-term use of high doses of statins can induce liver dysfunction.[9] Fortunately, traditional cultivation health methods, such as Tai Chi, yoga and Baduanjin, are popular with middle-aged and older people and are considered to cultivate the body and promote health.[10 11]

### Description of the intervention

Baduanjin, which originated in China over 1000 years ago, is a qigong practice comprising eight sections of gentle movements and relaxation postures. It is based on the traditional Chinese medical theory of qi.[12] Compared with other exercise therapies, Baduanjin is easy to master in a short time and has fewer physical demands.[13] Baduanjin exercises can help stretch limbs and muscles and regulate

the breath, which improves the coordination of the body, harmonises qi and blood and contributes to physical and mental well-being.[14] Baduanjin exercises are popular in the middle-aged and elderly population because regular practice helps maintain positive physical and mental states.[15 16]

Many clinical trials have shown that Baduanjin exercises can substantially lower levels of TC, triglyceride (TG) and LDL-C and can increase high-density lipoprotein cholesterol (HDL-C) levels. However, other studies have found that Baduanjin practice has no significant effects on blood lipid levels.[17] There have been no relevant systematic reviews and meta-analyses on the effects of Baduanjin on elevated blood lipid levels in middle-aged and elderly individuals. In addition, there is insufficient evidence to support the widespread use of Baduanjin. Consequently, an examination of this therapy's effect on elevated blood lipid levels is needed. In this study, we plan to conduct a systematic review and meta-analysis to evaluate current evidence on the effects of Baduanjin on elevated blood lipid levels in middle-aged and elderly participants.

## METHODS AND ANALYSIS
### Eligibility criteria
#### Study type
Regardless of the length of treatment, we will include all randomised controlled trials that evaluated the safety of Baduanjin exercises and their effect on elevated blood lipid levels in middle-aged and elderly people.

### Participants
We will include studies of middle-aged and elderly participants including the following groups: physically healthy individuals, those with hyperlipidaemia, cardiovascular diseases, stroke, hyperlipidaemia combined with hypertension (based on 2010 Chinese guidelines for the management of hypertension[18]), or hyperlipidaemia combined with diabetes (based on the WHO's definition of diabetes[19]).

### Interventions
Control group patients will comprise those receiving no treatment, routine treatment or other conventional exercise therapies such as jogging or walking. The intervention group will receive Baduanjin exercises of various duration and frequency based on routine regimens.

### Outcome measures
The primary outcome measures will comprise four blood lipid indexes: TC, TG, LDL-C and HDL-C.

The secondary outcome measures will be body mass index, waist-to-hip ratio, Apolipoprotein A1, Apolipoprotein B and adverse events.

### Search strategy and identification of studies
We will retrieve articles from the following databases: PubMed, Embase, Cochrane Library, Chinese National Knowledge Infrastructure, Wanfang Data, Chinese

**Table 1** Search strategy for the PubMed database

| Number | Search terms |
|--------|-------------|
| 1 | blood lipid |
| 2 | blood fats |
| 3 | hyperlipidemia |
| 4 | hyperlipaemia |
| 5 | hypercholesterolemia |
| 6 | hypertriglyceridemia |
| 7 | hyperlipoproteinemia |
| 8 | total cholesterol |
| 9 | triglycerides |
| 10 | low-density lipoprotein cholesterol |
| 11 | high-density lipoprotein cholesterol |
| 12 | dyslipidemia |
| 13 | or1-12 |
| 14 | Qigong |
| 15 | Baduanjin |
| 16 | Baduanjin exercise |
| 17 | eight section brocades |
| 18 | Or 14–18 |
| 19 | And 13–18 |

BioMedical Database and Chinese Science and Technology Periodicals Database. The publication period will be from inception to March 2017. The search terms will be 'blood lipid' OR 'blood fats' OR 'hyperlipidemia' OR 'hyperlipaemia' OR 'hypercholesterolemia' OR 'hypertriglyceridemia' OR 'hyperlipoproteinemia' OR 'total cholesterol' OR 'triglycerides' OR 'low-density lipoprotein cholesterol' OR 'high-density lipoprotein cholesterol' OR 'dyslipidemia' AND 'Qigong' OR 'Baduanjin' OR 'Baduanjin exercise' OR 'eight section brocades'. Chinese translations of these terms will be used for the Chinese databases. Table 1 shows the search strategy for PubMed.

Two reviewers (TL and CJ) will independently review the full texts of potential eligible studies. Discrepancies about inclusion in the meta-analysis will be discussed and settled by a third reviewer. A Preferred Reporting Items for Systematic Reviews and Meta-Analyses (PRISMA) flow chart will be produced to show the number of articles identified, screened, included and excluded, reasons for exclusion and to ascertain eligible studies. The study selection process will be described in a PRISMA flow chart (http://www.prisma-statement.org) (figure 1).

### Study selection and data extraction
We will exclude articles for which no data on blood lipid outcomes is presented, relevant information is unavailable or results are duplicated. Two reviewers (TL and CJ) will use the criteria described above to independently review titles and abstracts to select potential references and then

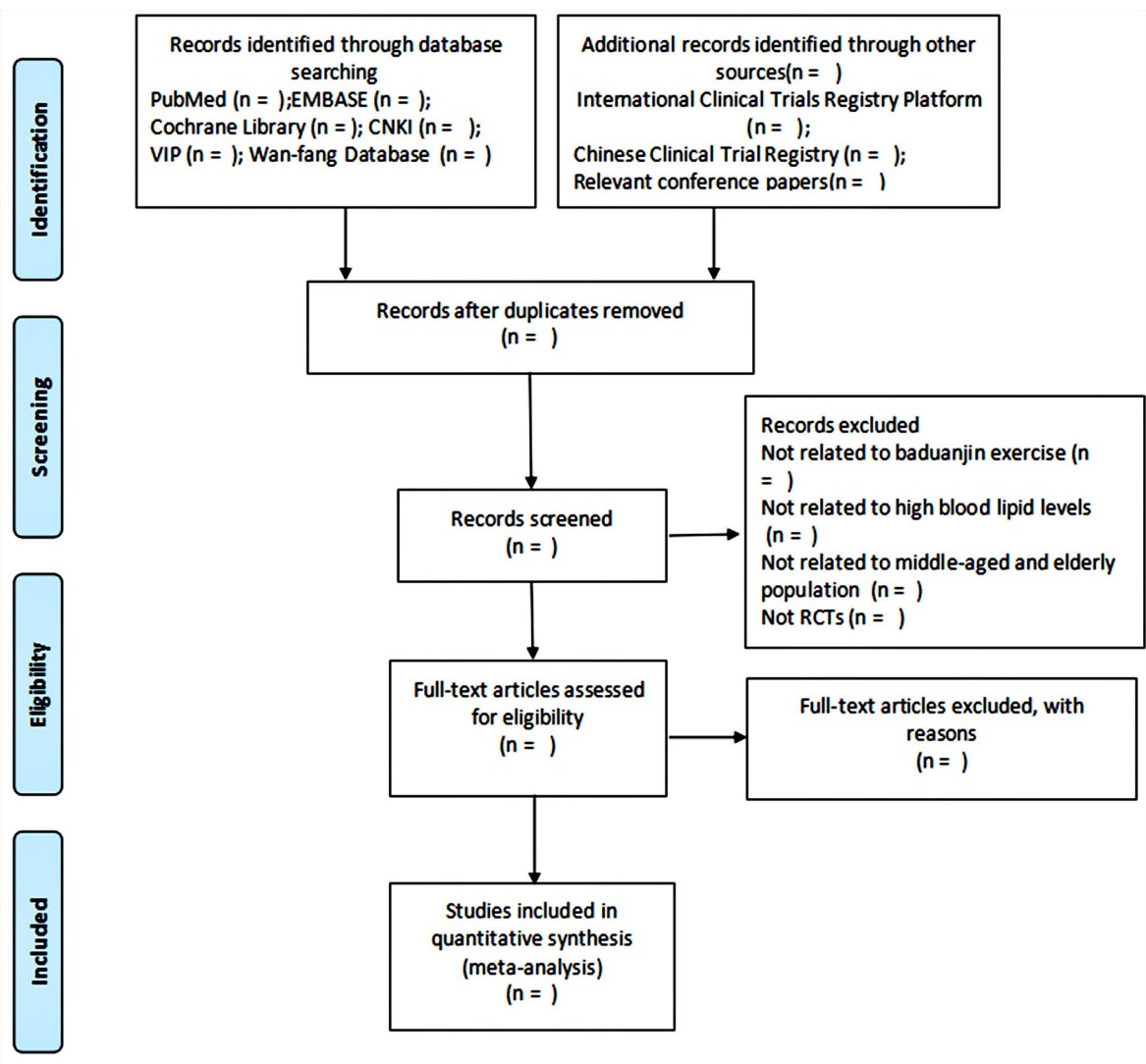

**Figure 1** Flow diagram of study selection process. CNKI, Chinese National Knowledge Infrastructure; VIP, Chinese Science and Technology Periodicals Database.

to scan all full articles for eligibility. Endnote V.X7 will be used to manage literature and remove duplications.

The reviewers will then extract data about the study population, intervention and outcome measures using a self-developed data extraction form. The data extraction form includes the following items: general information, trial characteristics, participants, interventions and outcomes. Disagreements will be resolved through consultation and if necessary, discrepancies will be discussed with a third author.

### Addressing missing data or unclear measurement scales

If necessary, we will contact authors of articles by email or telephone to request missing data or additional information about evaluation scales. If sufficient information cannot be obtained in this way, we will analyse the available data. We will take into account the potential impact of insufficient data on the review results (in the Discussion section).

### Risk of bias in included studies

Risk of bias for each included article will be evaluated according to the Cochrane Handbook for Systematic Reviews of Interventions. This recommends the assessment of several sources of bias, including random sequence generation, allocation concealment, blinding of outcome assessments, incomplete outcome data and selective outcome reporting. Bias among participants and investigators will not be considered because the Baduanjin exercises make blinding impossible. We will judge the risks as low, high or unclear (unclear or unknown risk of bias).

### Data synthesis and analysis

We will analyse the data using Stata V.13.0. We will calculate weighted mean differences and 95% CIs. The Q and $I^2$ test statistics will be used to assess the heterogeneity of included studies. For the Q statistic, $p<0.05$ will be considered as indicating significant differences. For the $I^2$ statistic, $I^2<25\%$ indicates no significant heterogeneity,

$I^2=25\%–50\%$ is considered moderate heterogeneity and $I^2>50\%$ indicates strong heterogeneity. We will use fixed effects models if there is no heterogeneity among studies and random effects models if there is heterogeneity.

### Additional analyses

We will perform sensitivity analysis, metaregression and subgroup-stratified analysis based on various study characteristics, such as study type, study location, sample size, study quality, adjustment (or not) for confounders, Baduanjin regimens and other relevant parameters (eg, diet and lifestyle) to explore potential sources of heterogeneity. If data extraction is insufficient, we will create a qualitative synthesis.

### Assessment of reporting biases

A funnel plot will be developed to evaluate reporting bias of the included studies. We will use Begg and Egger tests to assess funnel plot symmetry and will interpret values of p<0.1 as showing statistical significance (ie, publication bias).

### Quality of evidence

The Grading of Recommendations Assessment, Development and Evaluation approach will be used to evaluate the quality of evidence of the included studies. Reviewers will take into account limitations of the study, inconsistencies, indirect evidence, inaccuracies and publication bias. Four levels of evidence quality will be used: high, moderate, low or very low.

## ETHICS AND DISSEMINATION

Our aim is to publish this systematic review in a peer-reviewed journal. Our findings will provide information about the safety of Baduanjin exercises and their effect on blood lipid levels of middle-aged and elderly people. This review will not require ethical approval as there are no issues about participant privacy.

## DISCUSSION

Baduanjin is a form of qigong that has been practised for more than 1000 years in China. Baduanjin comprises slow, relaxing and systematic movements that are suitable for physically weak and elderly patients.[20] Although the movements are soft and slow, the physical and mental aspects of this practice can strengthen the body muscles and tendons and relax the ligaments.[21] Previous studies have indicated that Baduanjin can obviously increase plasma HDL-C concentrations and decrease plasma TC, TG and LDL-C concentrations in both healthy people and patients.[13] One study showed that qigong can inhibit cholesterol biosynthesis in one or more pathways to improve lipid synthesis (eg, by promoting TG transport and degradation, increasing TC scavenging ability and increasing TC ability to transport and reduce acetyl coenzyme A).[22] Research has also shown

that long-term exercise can activate levels of lipoprotein (a rate-limiting enzyme that causes TG decomposition and determines the amount of fatty acids in skeletal muscle), which accelerates lipid metabolism and improves blood lipid levels.[23]

Baduanjin could therefore be used to treat high blood lipid levels, but the efficacy of this exercise regimen compared with hypolipidaemic drugs is unknown. In addition, there is insufficient evidence that Baduanjin exercise can lower abnormal blood lipid levels. The purpose of this review is to systematically assess the effect of Baduanjin exercises on blood lipid levels in middle-aged and elderly individuals. We aim to use enough studies to ensure adequate power for the meta-analysis. We expect to find that Baduanjin exercises have a positive effect on primary prevention of high blood lipid levels in middle-aged and elderly individuals. In summary, this review will be the first to evaluate the impact of Baduanjin exercise on primary prevention of high-blood lipid levels in middle-aged and elderly individuals. The results of this review may help to establish a better approach to prevention of high-blood lipid levels in high-risk groups and to provide reliable evidence for its application.

**Contributors** WW and JW conceived the study and drafted the protocol. RP revised it. TL and CJ developed the search strategies, conducted data collection and analysed independently. All authors have approved the final manuscript.

**Funding** This study is supported by a project of the Administration of Traditional Medicine of Guangdong (grant no. 2015 LP01).

**Competing interests** None declared.

**Provenance and peer review** Not commissioned; externally peer reviewed.

**Data sharing statement** No additional data are available.

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
