## [Reviewer comments · BMJ Open]

ARTICLE DETAILS

TITLE (PROVISIONAL)	Effect of Baduanjin exercises on elevated blood lipid levels of middle-aged and elderly individuals: Protocol for a systematic review and meta-analysis of randomized controlled trials
AUTHORS	Wen, Junmao; Lin, Tong; Jiang, Chenguang; Peng, Rui; Wu, Wei

VERSION 1 - REVIEW

REVIEWER	Nain-Feng Chu School of Public Health, National Defense Medical Center, Taipei, Taiwan Department of Medical Education and Research, Kaohsiung Veterans General Hospital, Kaosgiung, Taiwan
REVIEW RETURNED	14-May-2017

GENERAL COMMENTS	This is a protocol to evaluate the efficacy and safety of Baduanjin on blood lipids among middle-aged and elderly population. The materials and methods are quite straight forward and easy conducting, however, there are some points need further discussion. 1. For Baduanjin – it is very difficult to conduct randomized controlled trial; furthermore, some dietary and lifestyle factors may be associated with blood lipid profiles – but did not considered in this protocol.2. Patients with hypertension and/or diabetes are included in this study – however, these patients may be associated with dyslipidemia and should be analyzed very carefully – either due to exercise programs or medications.3. Patients of control groups includes no treatment, routine treatment and other exercise therapy – there are so varying for the degree of physical activities and potential confounding should be considered.4. In this protocol, the duration and frequency of Baduanjin therapy is not considered – neither for the age or history of disease.5. There are some typos for total cholesterol or triglyceride in the manuscript.
--

REVIEWER	Yonghong Zhang Soochow University
REVIEW RETURNED	15-May-2017

GENERAL COMMENTS	The authors prepare to evaluate the impact of Baduanjin exercise on primary prevention of high blood lipid level in middle-aged and elderly population and the results of this meta-analysis will help to establish the better approach for preventing blood lipid level in high risk groups and provide reliable evidence for its application. The following are some comments concerning the study. 1. The aim of this study is interesting.
---

	2. Why authors do not include patients with hyperlipidemia, cardiovascular disease and stroke in the section of “types of participants”. 3. The search terms should include total cholesterol, triglycerides, low-density lipoprotein cholesterol, high-density lipoprotein cholesterol and dyslipidemia. 4. What is the “other source” in the “additional records identified through other sources” in Figure 1.
--	--

VERSION 1 – AUTHOR RESPONSE

Replies to Reviewer 1

1. For Baduanjin – it is very difficult to conduct randomized controlled trial; furthermore, some dietary and lifestyle factors may be associated with blood lipid profiles – but did not considered in this protocol.

Answer:(1) In fact, we agree with the opinion that it is difficult to conduct randomized controlled trial for Baduanjin. However, some randomized controlled trials still exist, though there are some methodological flaws. In addition, we also search some papers about the meta-analysis on Baduanjin or other exercise therapies of Chinese medicine (for example, Tai ji). Therefore ,we will conduct a meta-analysis to explore the lowering effect on elevated blood lipid levels according to Baduanjin, and we will make a full consideration on the methodological flaws.

(2)Dietary (for example, high-calorie, high-fat diets) and lifestyle factors (such as smoking, excessive drinking and burning mid-night oil) has been added in the revised version (page 9, paragraph2, line5)

2. Patients with hypertension and/or diabetes are included in this study – however, these patients may be associated with dyslipidemia and should be analyzed very carefully – either due to exercise programs or medications.

Answer:(1) Dyslipidemias are common in patients with hypertension and diabetes mellitus, therefore, patients diagnosed with hypertension or diabetes mellitus combined with dyslipidemias will be included. Correction has been made in the revised version (page 5and 6, the section of “participants”)

(2)Exercise programs, antidiabetic and anti-hypertensive drugs will have a impact on the effect of lowering blood lipid levels. Therefore, we will focus on the exercise programs or medications and make a subgroup analysis if data enough.

(3)Patients of control groups includes no treatment, routine treatment and other exercise therapy – there are so varying for the degree of physical activities and potential confounding should be considered.

Answer: our objective is to explore which regimens of baduanjin exercise is better for patients with high blood lipid levels, that’s the reason why we varying for the degree of physical activities. And we will conduct a sensitivity analysis of Baduanjin regimens (page 9, paragraph2).

4. In this protocol, the duration and frequency of Baduanjin therapy is not considered – neither for the age or history of disease.

Answer: the duration, frequency, age and history of disease will be presented in the basic characteristics of included studies and we will make a table about it when we conduct a meta-analysis. Otherwise, we will focus on the potential sources of heterogeneity caused by age or history of disease .

Replies to Reviewer 2

1.The aim of this study is interesting.

Answer: Thanks.

2. Why authors do not include patients with hyperlipidemia, cardiovascular disease and stroke in the

section of “types of participants”.

Answer: Correction has been made in the revised version (page 5 and 6, the section of “types of participants”)

3. The search terms should include total cholesterol, triglycerides, low-density lipoprotein cholesterol, high-density lipoprotein cholesterol and dyslipidemia.

Answer: Correction has been made in the revised version (page 7, the section of “ search strategy and identification of studies”)

4. What is the “other source” in the “additional records identified through other sources” in Figure 1.

Answer: “other source” include International Clinical Trials Registry Platform, Chinese Clinical Trial Registry and Relevant conference papers. Correction has been made in the Figure 1.